# Practices and Barriers in Implementing the Low FODMAP Diet for Irritable Bowel Syndrome Among Malaysian Dietitians: A Qualitative Study

**DOI:** 10.3390/nu16213596

**Published:** 2024-10-23

**Authors:** Tham Jin Ke, Mohd Jamil Sameeha, Kewin Tien Ho Siah, Putri Balqish Qistina Binti Jeffri, Noor Athierah Binti Idrus, Shanthi Krishnasamy

**Affiliations:** 1Dietetics Program, Faculty of Health Sciences, Universiti Kebangsaan Malaysia, Jalan Raja Muda Abdul Aziz, Kuala Lumpur 50300, Malaysia; p114215@siswa.ukm.edu.my (T.J.K.);; 2Centre for Community Health Studies (ReaCH), Faculty of Health Sciences, Universiti Kebangsaan Malaysia, Jalan Raja Muda Abd Aziz, Kuala Lumpur 50300, Malaysia; sameeha@ukm.edu.my; 3Department of Medicine, Yong Loo Lin School of Medicine, National University of Singapore, Singapore 117597, Singapore; kewinsiah@nus.edu.sg; 4Centre for Diagnostic, Therapeutic and Investigative Studies (CODTIS), Faculty of Health Sciences, Universiti Kebangsaan Malaysia, Jalan Raja Muda Abdul Aziz, Kuala Lumpur 50300, Malaysia

**Keywords:** Asian, irritable bowel syndrome, low FODMAP diet, FODMAPs, qualitative study

## Abstract

The low fermentable oligo-, di-, mono-saccharides and polyols (FODMAP) diet (LFD) is a second-line dietary intervention for irritable bowel syndrome (IBS) patients, involving FODMAP restriction, reintroduction, and personalization, and it needs to be delivered by dietitians. However, the application of this diet among Malaysian IBS patients is not well understood. This study aimed to explore the practices and barriers in delivering the LFD among Malaysia dietitians. Semi-structured qualitative interviews were conducted online with practicing dietitians until the data reached saturation. All the interview sessions were audio recorded and transcribed verbatim. Thematic analysis was used to analyze the data. Eleven dietitians were interviewed, with 36.4% (*n* = 4) having more than 10 years of experience. The following four themes regarding their practices emerged: 1. dietary advice on FODMAP restriction; 2. duration of FODMAP restriction phase; 3. references used to get information about FODMAPs, and 4. strategies on reintroduction. Meanwhile, the following seven barriers were identified: 1. lack of culturally relevant educational materials; 2. limited knowledge about the LFD; 3. inadequate formal training among dietitians; 4. lack of integration in multi-disciplinary care; 5. low health literacy of patients; 6. low compliance rate among patients, and 7. restrictions for certain populations. LFD implementation in Malaysia is not standardized as only experienced dietitians can provide dietary evidence-based advice. Lack of training and culturally specific resources are some of the main barriers that were identified to be limiting the implementation of the diet. Therefore, there is a need for training programs and resource development to support Malaysian dietitians in managing IBS patients.

## 1. Introduction

Irritable bowel syndrome (IBS) is a chronic functional bowel disorder characterized by recurrent abdominal pain and altered bowel habit without any organic causes [1]. In the last two decades, the efficacy of the low fermentable oligo-, di-, mono-saccharides and polyols (FODMAP) diet (LFD) in alleviating IBS symptoms, including abdominal pain, bloating, distension and stool consistency, especially when delivered in a structured education format by trained dietitians has been well demonstrated [2,3,4]. As such, it is now recommended as a second-line therapy for IBS patients with persisting symptoms by dietetic and gastroenterology organizations worldwide [5,6,7,8]. However, this dietary approach is relatively new in Southeast Asia, where clinical studies of LFD implementation in the regions are still lacking. To date, there has only been one observational pilot study conducted in Malaysia. Although some patients showed symptom improvement, the sample size was relatively small and compliance was low among the patients [9].

The traditional Malaysian diet is very diverse and reflects its multi-ethnic population. Malay, Chinese, Indian, and East Malaysian foods make up the Malaysian diet, which is high in FODMAPs, mainly from the onions, garlic, and shallots used in cooking [10], tropical fruits that are high in fructose [11], as well as refined wheat products [12]. However, the FODMAP intake among Malaysians is still unknown. Therefore, FODMAP restriction may be quite challenging without baseline data on FODMAP consumption among Malaysians.

According to a technical report in 2019 [13], there are approximately 446 dietitians practicing in Malaysia. To become a qualified dietitian in Malaysia, one must complete one of following three recognized pathways: (i) a 4-year undergraduate degree in Dietetics, (ii) a postgraduate degree such as a Master’s in Dietetics, or (iii) a Postgraduate Diploma in Dietetics from a recognized institution, either in Malaysia or abroad [14]. In addition, the Malaysian Qualifications Agency requires dietitians to complete 784 to 1200 h of internship training to be eligible for practice [15].

While these pathways ensure that dietitians are equipped with the foundational skills, there are no pathways for specialized dietitians. Therefore, the number of dietitians seeing patients with gastrointestinal disorders, specifically IBS, is limited. This limitation presents a challenge, as delivering the LFD effectively requires specific knowledge and training. Dietitian-led interventions are crucial for personalized patient care and improved compliance in a real-life setting. Dietary implementation by a non-dietitian or through an app may compromise efficacy and patients’ compliance [16,17]. In addition, nutritional inadequacy risk may arise, especially for those who apply the LFD without professional guidance [18,19]. To achieve this, dietary intervention needs to be delivered and supervised by dietitians trained specifically in delivering the LFD.

However, research on the LFD delivery practices and barriers faced by dietitians is scarce in both the Western and Asian settings. To address this gap, the aim of this study was to explore practices and barriers faced by Malaysian dietitians when delivering the diet for IBS patients through a semi-structured interview.

## 2. Materials and Methods

### 2.1. Study Design

This was a qualitative study with a phenomenological design focusing on dietitians’ experiences and barriers in delivering the LFD for IBS patients in Malaysia. Ethical approval was obtained from Secretariat of Research Ethics Committee, Univerisiti Kebangsaan Malaysia (UKM PPI/111/8/JEP-2022-193). Due to the COVID-19 pandemic, interviews were conducted via virtual platform, either the Zoom or Microsoft Teams video conferencing platforms, from July to September 2022. Although the COVID-19 pandemic had transitioned to an endemic phase during the study period, we chose to continue conducting interviews virtually for several reasons. There were still ongoing safety concerns and varying levels of comfort with in-person interactions among the participants. Additionally, virtual platforms offered greater flexibility and convenience, making it easier to schedule interviews and reducing travel time. This approach also allowed us to include participants from various locations without logistical issues.

### 2.2. Study Population

A purposive sampling technique was used to recruit practicing dietitians with a minimum of one year of experience who have counseled IBS patients and used this diet before. This duration of experience was considered sufficient as, within the Malaysian context, specialized gastrointestinal (GI) dietitians are not available, and most dietitians tend to rotate between specialties, particularly within private settings, after one year of practicing. Additionally, dietitians often encounter IBS patients in their outpatient settings due to the lack of dedicated GI clinics in Malaysia. Retired dietitians were excluded from this study. A recruitment poster with the study information and registration QR code was created and distributed through various channels, including the Malaysian Dietitians Association (MDA), heads of the hospitals and social media platforms such as Instagram (Version 345), Facebook (Version 475), and WhatsApp (Version 2.24).

### 2.3. Study Procedures

#### 2.3.1. Development of Interview Guide

An interview guide with a sequence of open-ended questions was developed based on the research aims pertaining to the LFD to facilitate and standardize the interview sessions (Appendix A). Some key papers informing the guide included publications related to the implementation of the three phases of the LFD [20,21,22,23,24], in addition to guidelines from National Institute for Health and Care Excellence (NICE) [8], the American College of Gastroenterology (ACG) [6], and the British Dietetic Association (BDA) [25].

The guide was refined through discussions among researchers with expertise in qualitative study and FODMAPs. The interview began with some baseline questions about their working experiences and strategies in advising the LFD. For example, the duration of the restriction phase, how they monitor their patient’s symptoms, how to help patients identify high FODMAP foods or drinks, and where to find information about LFD. The barrier section had two important questions that focused on the challenges encountered in delivering LFD and the strategies to overcome them.

#### 2.3.2. Pilot Test

Before conducting the actual interviews, the questions were piloted by interviewing two researchers in related fields and one practicing dietitian who works in a clinical setting to ensure the questions were relevant and understandable. Based on the feedback from the pilot interviews, several modifications were made to simplify and enhance participant understanding. For example, the question “Can you let us know the barriers you faced when you implemented the diet in all three phases?” was refined to “What difficulties do you face in implementing the diet?” Similarly, “Are you aware of the phases in the low FODMAP diet? was changed to “Can you share with me what you know about the low FODMAP diet?”

#### 2.3.3. Data Collection

The semi-structured interviews were conducted in English and were video recorded. There is no specific formula for calculating sample size in qualitative research. In the present study, data collection was continued until the data reached the “saturation” point, a point where new codes or themes cannot be discovered and no new information was found [26,27]. Upon the agreement of dietitians to participate in the interviews, a participant information sheet, a Google Form link containing socio-demographic information, and a consent form, were sent to study participants via email or WhatsApp prior to the interview sessions. The sociodemographic form included basic information such as name, gender, age, years of experience, workplace, and frequency of counseling IBS patients.

The interview session began with a warm welcome and a brief introduction of the researcher, then an overview of the research topic and objectives, an outline of the ground rules, the estimated duration of discussion, and an assurance of confidentiality were provided. Participants were encouraged to share their experiences and opinions throughout the interview sessions and probed further when the answers were not clear to capture in-depth information about the subject or topic. At the end of each interview, participants were given a summary of findings from their interview to confirm that the researcher’s understanding was consistent with the participant’s context. Each participant received an inconvenience allowance as a token of appreciation for their participation in the study.

Data analysis was conducted concurrently with data collection, whereby emerging codes and themes were continuously compared with previously collected data. The data saturation was achieved at the 11th dietitian, as no new themes emerged in the last three participants.

### 2.4. Trustworthiness and Reflexivity

In this study, several strategies were employed to enhance the trustworthiness and rigor of the interview findings. Firstly, transferability was addressed through comprehensive descriptions of the study’s setting, participant criteria, data collection, and analysis procedures outlined in previous sections [27,28]. All recordings, transcripts, and field notes were meticulously documented to ensure its dependability [29]. Peer debriefing was conducted through several meetings with experts in qualitative research, IBS, and FODMAPs to discuss various aspects of the research, including research methods, data analysis and interpretation, to ensure the robustness of the findings [29].

Reflexivity was conscientiously integrated to ensure transparency. Despite prior knowledge of FODMAPs and IBS, the researcher avoided providing cues or hints that could lead participants towards specific answers and adopted a neutral stance while listening to their perspectives without imposing any personal views. Additionally, interview questions were carefully crafted to be open-ended and non-directive, allowing the authentic expression of thoughts and experiences of participants.

### 2.5. Data Analysis

Interview data were analyzed using the inductive thematic analysis approach, which is a data-driven process, as the codes were derived from the data [30,31]. Following each interview session, the recordings were transcribed manually using Microsoft Word (Microsoft 365 MSO, version number 2408). The recordings and transcripts were reviewed multiple times to gain familiarity with the data and identify the major organizing ideas. At the same time, detailed notes and key concepts were made to ensure the comprehensive coverage of relevant details for the study.

Subsequently, the related data or data with similar meaning were organized into preliminary codes to facilitate the researcher’s understanding from each participant’s perspective. In this stage, a systematic approach was employed; herein, a table was created using Microsoft Word to organize the important quotes. Each quote was assigned a short phrase or term derived directly from the data to ensure that the codes were closely tied to the data [32]. Additionally, coding cycles were employed to revise and refine the codes. Codes were clustered together based on their similarity and regularity, giving rise to patterns that allow for the comparison and analysis of connections between them. Next, the relevant codes were further grouped into potential themes that captured key concepts. Once the main themes were identified, several reviews and discussions were conducted with the research team to ensure the content of these themes was not conflicting and was appropriate for the subject matter [30] until a mutual agreement was achieved.

## 3. Results

Table 1 shows the demographic characteristics of the 11 participants who completed the interviews. The majority of participants were female (*n* = 9, 81.8%) and aged between 30 and 39 years old (*n* = 6, 54.5%). Only one-third (*n* = 4, 36.4%) had over ten years of dietetics experience, while most worked in private hospitals (*n* = 6, 54.5%). On average, the interview sessions lasted 20 to 30 min. In this study, all participating dietitians were from West Malaysia. The dietitians are identified as D1–D11 in the following section to reference their responses.

To better understand how dietitians currently implement the LFD, participants were asked several questions about their practices during each phase of the diet. The following four key themes were generated from the thematic analysis process: (i) dietary advice on the restriction phase, (ii) duration of FODMAP restriction phase, (iii) references used to get information about FODMAPs, and (iv) strategies on reintroduction (Table 2).

A few barriers were reported in detail by the participants and were classified into the following two categories: (i) challenges related to current practice and (ii) challenges in patient management (Table 3). In the area of challenges related to current practice, four main themes emerged as follows: (i) lack of culturally relevant educational materials, (ii) limited knowledge about the LFD, (iii) inadequate formal training among dietitians, and (iv) lack of integration in multi-disciplinary care. On the other hand, challenges in patient management included (i) low health literacy of patients, (ii) low compliance rate among patients, and (iii) restrictions for certain populations.

## 4. Discussion

To the best of the author’s knowledge, this is the first qualitative study that has explored dietitians’ practices in LFD delivery and the barriers faced by them in delivering the diet to patients in Malaysia. It highlighted different perspectives regarding the important role of dietitians in medical nutrition therapy, as well as the need for skills training and resources in addressing the diet to patients and addressed the current gaps in the real-world day-to-day practice of dietitians.

### 4.1. Dietitians’ Practices in Low FODMAP Diet Delivery

One of the key findings was the variation in the recommended duration of the FODMAP restriction, which deviated from the guidelines. Although the clinical guidelines and research studies generally suggest a restriction phase of two to six weeks [25,33,34], some participants in this study reported durations as short as less than two weeks or as long as more than three months. One of the factors that contributed to these variations is a potential lack of comprehensive awareness about the recommended duration of the diet among some dietitians. During the interview, researchers found that some of the dietitians were not fully aware of the guidelines thus suggesting a restriction period of less than two weeks. Another possible reason is appointment availability. Since patients’ appointments are typically scheduled by doctors and due to the constrained slot availability, especially in public hospitals, patients may be unable to revisit within three months. This was mentioned by two participants working in public hospitals.

A minimum two-week FODMAP restriction reduced the overall symptoms in patients in several randomized controlled trials all over the world [35,36,37]. An overly short restriction may be ineffective for adequately controlling IBS symptoms and identifying the trigger foods [38]. On the contrary, prolonged restrictions of more than three months may lead to several issues such as nutrient inadequacies, gut microbiome alterations, and reduced patient compliance to the diet. For instance, a cross-sectional study of over 3000 patients in Iran found that patients who followed the diet strictly had a significantly lower dietary intake of all foods and nutrients, including calcium, magnesium, vitamin C, folate, and riboflavin [19]. Similarly, a recent study comparing the LFD and modified NICE diets at four weeks demonstrated a reduction in the mean intake of several micronutrients, including thiamine, riboflavin, calcium, and sodium [39]. Although only significant changes in riboflavin were noted after adjusting for energy intake, this underscores the risk of malnutrition if they restrict high FODMAP foods longer than the necessary period.

These findings highlight a critical risk—even a four-week restriction phase, when followed strictly, can lead to nutrient deficiencies, especially in calcium which is vital for bone health and various physiological functions, due to the restriction of milk and dairy products that are rich in FODMAPs. If this phase is unnecessarily prolonged, the risk of malnutrition may increase, exacerbating the concerns raised in previous research.

It is also noted that the LFD may negatively impact intestinal microbiome with FODMAP restriction, due to the decreased intake of fiber-, prebiotic-, and probiotic-rich foods. Studies have revealed a significant reduction in total microbial amount and abundances of health beneficial Bifidobacteria [40,41], after restricting high FODMAP foods. One randomized controlled trial study has found a decrease in the total bacterial count within just a week of a low FODMAP diet [42]. Although most trials were short-term, these results indicate that the prolonged restriction of FODMAPs beyond the suggested period could have a detrimental effect on the gut microbiome. Therefore, FODMAP restriction is recommended only for short-term use. These inconsistencies in the duration suggested to patients indicate that the implementation was not based on evidence-based practice or guidelines, and there is a need for a standardized practice among dietitians.

Although most participants were familiar with the LFD, many were unfamiliar with the process involved during the reintroduction phase or how to implement this phase. Only one of them was able to explain the whole process correctly according to the guidelines, suggesting a need for enhanced training and education in this area. The reintroduction phase involves systematically reintroducing FODMAPs back into the diet in order to identify the individual trigger foods and increase the dietary diversity and nutritional status of the patients instead of unnecessarily omitting foods in the long term [22].

During the reintroduction phase, patients need to continue a strict LFD and only introduce one group of FODMAPs at a time, starting with the minimum portion size on day one. This amount is then increased gradually on day two and day three. Patients will then record the foods eaten and monitor their symptoms in response closely in terms of type and severity during the three-day reintroduction period. After the three-day washout period, the next FODMAP category can then be reintroduced and tested in the same manner. If the food is well tolerated and there is no significant impact on symptoms, it can likely be included regularly in the patient’s diet. However, if mild symptoms occur but are still manageable, it should be consumed in smaller servings or omitted until the patient feels comfortable. If symptoms are severe or substantially increase during the three-day reintroduction period, patients should discontinue and return to the LFD for another three days until no symptoms occur before reintroducing a new food [21,43]. This systematic reintroduction is a key step for transitioning from the initial restriction phase to form a personalized LFD for long-term symptom management.

A recent prospective study conducted in 2022 found that patients who underwent a structured FODMAP restriction, reintroduction, and personalization process maintained symptom relief compared to the baseline and normalized microbiota composition changes seen after short-term restriction, during a long-term follow-up of, in this case, one year [44]. This emphasizes the importance of comprehensive dietitian guidance, standardized guidelines, as well as the appropriate resources to facilitate the delivery of the LFD. However, our findings revealed that many dietitians resort to creating their own written resources for patients, and these are either based on their personal knowledge or through discussion with their senior colleagues or students. Further probing revealed that these resources are not evidence-based as most of the information was obtained online through various websites. Reliance on non-evidence-based resources raises significant concerns about the quality and accuracy of the information provided to patients, leading to suboptimal outcomes. Patients may experience difficulties in understanding and implementing the diet, potentially impacting compliance.

### 4.2. Barriers in Low FODMAP Diet Delivery

In addition to this, one of the main challenges reported by participants was the lack of culturally appropriate educational materials or resources. This is in line with previous reviews from Southeast Asian countries and trials highlighting suboptimal educational resources as a barrier, with a dependence on Western food lists and apps despite different regional diets [45,46]. While reliable educational materials exist, such as “Monash University Low FODMAP: The Cookbook” priced at USD $44.95 and the King’s College London education booklets [47], which are only accessible to registered dietitians at a cost of £45, their costs are high and less applicable in Malaysia’s cultural context. For instance, many of the low FODMAP ingredients featured in the recipes and shopping lists, such as corn tortilla, gluten-free bread or flour, lactose-free yogurt, and others, are not readily available in local grocery shops in Malaysia. Additionally, common condiments like chili sauce, sambal, and oyster sauce often contain high FODMAP ingredients such as onions and garlic, further challenging the practice of the diet.

Given the increasing prevalent trend of dining out in Malaysia [12], a culturally relevant materials, such as a booklet, is of the utmost importance for patients to make informed choices for FODMAP-friendly foods when dining out and even preparing home-cooked meals with the modified recipes. A lack of culturally appropriate resources may result in the over-restriction of FODMAPs due to incomplete or outdated recommendations. If resources are limited, this may drive patients to seek inaccurate information elsewhere [17]. Moreover, the available information may be inconsistent or contradictory between different resources [48,49], making the patients confused. Therefore, there is a need for guides tailored to the cultural context, providing suitable local food substitutions, guidelines on food label reading, and portion sizes to facilitate patient understanding and compliance. In addition, resources should also be regularly updated as new FODMAP composition data and clinical evidence emerge.

Participants in this study also mentioned that they rarely received an IBS patient referral from physicians or gastroenterologists, aligning with previous findings that physicians or gastroenterologists often provide incomplete low FODMAP advice themselves and yet infrequently refer patients to dietitians [17,50]. This low referral rate can be attributed to several factors. Firstly, the role of functional gastroenterologists is limited, and not all hospitals offer GI services. A participant also mentioned that doctors primarily rely on medication and consider diet as a last resort. Secondly, there is a lack of dedicated dietetic services and specialized clinics in hospitals, hence gastroenterologist or physician often provide dietary advice to patients during their consultations. Thirdly, dietetic service fees are not covered by insurance, further contributing to the low referral rate especially in private hospitals. However, delivery of the diet by doctors deviates from best practice as established by several guidelines [5,6], which recommend that it should be delivered by a trained dietitian to ensure optimal education and support, as this is not a one-size-fits-all restrictive diet. A collaborative, multidisciplinary care approach including a trained dietitian is advocated to provide optimal support for patients in comprehending and implementing this challenging diet.

Another significant barrier is the limited dietitian knowledge and lack of formal training opportunities for dietitians in Malaysia. The LFD is not comprehensively covered in undergraduate or postgraduate dietetic curriculum. Students are only given a brief introduction to the diet, often lacking practical implementation guidance in general dietetic textbooks, which do not cover the diet’s application and practical tips in-depth [51]. Consequently, students may gain theoretical knowledge in the classroom but lack practical experience, especially in GI nutrition as not all hospitals have specialized gastroenterology unit. Entry-level dietitians often do not engage with GI patients until later in their careers, typically during rotations in outpatient settings. Without sufficient confidence, dietitians are less likely to incorporate the LFD as part of their practice, which requires specialized skills in food composition, elimination protocols, and personalized reintroduction. Thus far, evidence for the successful implementation of the LFD and symptom relief has predominantly involved dietitian-led advice [36,44,52]. In this case, enhanced access to webinars, workshops, and accredited continuing education led by experienced providers, which are currently scarce in Malaysia, is critical for equipping dietitians with the skills and confidence required in delivering evidence-based LFD advice and guiding patients through this diet successfully. Currently, no dedicated LFD courses are offered in Malaysia, whereas only two countries, the United Kingdom [53] and Australia [54], offer specialized training programs that cover all the stages of implementation with practical tips. This gap in formal training contributes to the limited knowledge and application of the LFD among Malaysian dietitians.

Training should be made available at a low cost and should include assessments of competency before and after training to ensure that dietitians remain up to date with the latest evidence-based interventions. This effort should not be limited to the national level but should extend to regional levels as well. Collaboration with professional associations, including MDA, as well as private and public hospitals or universities could facilitate the establishment of specialized dietitian service clinics and provide ongoing education to all dietitians. By investing in this professional development, it provides a supportive environment for continuous learning and improves the quality of patient care and the nutrition care practices of dietitians.

Patients’ attitudes and behaviors towards the LFD are also a major challenge, with many struggling to comply with the diet. It is important to note that the low FODMAP diet may not be suitable for all patients, and adherence depends greatly on their motivation levels. The readiness to change model can be useful in understanding this aspect [55]. If a patient is not sufficiently motivated, they should first be given first-line dietary advice and then gradually introduced to the low FODMAP diet, or they can start with a “FODMAP-gentle” approach, which involves restricting a limited number of foods that are highly concentrated sources of FODMAPs and/or restricting suspected FODMAP foods that trigger symptoms [56]. This approach ensures that the diet is introduced in a manner that aligns with the patient’s readiness and motivation levels. Moreover, the rising trend of eating out in Malaysia poses additional barriers, as studies show over 60% of Malaysians eat at least one meal daily outside the home [12,57]. This growing reliance on dining out makes adherence difficult, as typical Malaysian cuisines often contain high FODMAP ingredients like garlic, onions, and coconut milk. The limited availability of suitable low FODMAP options in restaurants or cafeterias further impacts patients’ motivation to change and commit.

Apart from this, inadequate health literacy among patients is also perceived as a challenge in delivering the LFD which could negatively impact adherence, which is a key factor for successful IBS symptom management [46,58]. Participants mentioned that patients with low and medium education levels struggled to comprehend the concept and importance of the diet, along with recording the food intake in food diaries. A recent national health literacy survey found that 28% of Malaysian adults had inadequate health literacy, particularly those with less education, income, and older age [59]. The concept of FODMAPs and the need for dietary restriction or reintroduction is complex and may be difficult for patients with limited health literacy to grasp fully. Foreign resources assume a higher level of baseline nutrition knowledge and need the adequate adaptation to the Malaysian context. This underscores the need for tailored Malaysian educational materials using clear simple language, visual aids, and practical food label reading examples to match varying literacy levels.

### 4.3. Strengths and Limitations

This study has limitations in terms of sample size and generalizability. Although this qualitative study has reached saturation with 11 dietitians as participants, the findings may not be fully representative of the perspectives of all dietitians in Malaysia. A notable limitation is the small number of male dietitians and the absence of participants from East Malaysia in the sample, which may limit the diversity of the perspectives on professional practices. Despite these limitations, this study provides valuable initial insights into the real-world challenges dietitians face in implementing the LFD in Malaysia. Future studies should aim to include a more balanced gender representation to ensure a more comprehensive understanding of how gender may impact dietetic practice and challenges associated with the low FODMAP diet. Furthermore, future studies should assess the suitability, efficacy, and safety of this diet specifically for Malaysian IBS patients and develop educational materials tailored to Malaysia’s diverse ethnic populations and food options to enhance the outcomes of this diet.

## 5. Conclusions

In conclusion, this study provides insights into the current practices and barriers faced by dietitians in providing the LFD to IBS patients in Malaysia. Although the LFD is implemented as a second-line intervention therapy in Malaysia, the practice among dietitians is not standardized, and most dietitians were not able to provide dietary advice based on evidence-based practice. Lack of standardization in the practices and culturally specific resources, limited knowledge about the LFD, and inadequate training among dietitians are some of the main barriers that were identified to be limiting the implementation of the diet. Therefore, there is a need for training programs and resource development to support dietitians in managing IBS patients in Malaysia.

## Figures and Tables

**Table 1 nutrients-16-03596-t001:** Sociodemographic characteristics of participants.

Characteristics	Number of Participants (*n* = 11)	Percentages (%)
Gender		
Male	2	18.2
Female	9	81.8
Age (years)		
20–29	3	27.3
30–39	6	54.5
40–49	2	18.2
Practice settings		
Public	5	45.5
Private	6	54.5
Years of practice		
1–5	4	36.4
6–10	3	27.3
More than 10 years	4	36.4

**Table 2 nutrients-16-03596-t002:** Themes, codes, and quotes on dietitian practices in low FODMAP Diet delivery.

Themes	Codes	Quotes	Number of Mentions (%)
Dietary advice on FODMAPs * restriction	Avoid high FODMAPs * foods that may cause symptoms	*-“They need to restrict the high FODMAPs * containing food and substitute with the low FODMAPs *” (D1)* *“I will go through the foods that are either in the high FODMAPs * list or not included in the list with them, whereby patient need to follow the list strictly according to the amount and type of food that is allowed.” (D2)* *-“In first phase, they need to identify and avoid those foods that will cause symptoms” (D5)*	6 (54.5)
	Reduce eating out frequency	*-“ I usually tell my patients that if they really want to see the difference during this whole programme is that they will need to minimise eating out and prepare food from home” (D2)* *-“I asked them try to avoid as much as they to eat outside.”(D4)* *-“It can be challenging to go through each ingredient especially when eating outside. I will ask them to minimize eating out. We need to explain to them the disadvantages and advantages of preparing and eating food themselves.” (D11)*	5 (45.5)
	Guidance on reading labels	*-“I will teach them how to google search or how to buy product that is considered as FODMAPs * friendly” (D1)*	2 (18.2)
Duration of the FODMAPs * restriction phase	Less than two weeks	*-“If I’m not mistaken, less than two weeks”(D9)*	1 (9)
	Two to six weeks	*-“It will be around two to six weeks for this phase.” (D2)* *-“It’s about six weeks”(D7)* *-“ I ask them to avoid about two to four weeks ”(D10)*	6 (54.5)
	More than six weeks	*-“If I’m not mistaken, it’s about three to eight weeks or four to eight weeks” (D4)* *-“They need to restrict for at least for three months” (D3)* *-“Usually I just said avoid the gas-producing foods for two months”(D8)*	4 (36.4)
Reference used to get information about FODMAPs *	Self-developed resources	*-“I couldn’t recall when the resource was because it was hand down from my seniors in our hospitals” (D3)* *-“The FODMAPs * pamphlet that I am using was created by my student” (D5)* *-“Actually I created my own slides on FODMAPs *, then I just snapshot my slides and send it to them (patients) ” (D6)*	7 (63.6)
	Foreign websites and application	*-“My checklist is from Monash University and King’s College London (KCL)” (D1)* *-“I read from Monash university website, and I get the list of food from there.” (D4)* *-“In my hospital, we still don’t have pamphlets for FODMAPs *… I typically write down the notes and offer some website from the internet to them.”(D8)* *-“I will encourage them to download Monash apps.” (D11)*	4 (36.4)
Strategies on reintroduction	Reintroduce foods individually for three days and have a few of gap days before trying another new food	*-”Introduce one type of food for about three days and have gap days in between for each type of food group.” (D1)*	2 (18.2)
	Reintroduce foods individually for one week	*-“Introduce bit by bit but small portions for a week.” (D3)* *-“Reintroduce one kind of food first for one week. If there are no symptoms, they can add another food” (D7)*	2 (18.2)
	Recommendations for gradual reintroduction without durations	*-“They can reintroduce back the food but have to introduce slowly, one type of food at one time.” (D4)* *-“They can gradually reintroduce these foods and assess their tolerance levels through experimentation.” (D8)*	5 (45.5)

* FODMAPs: Fermentable oligo-, di-, mono-saccharides and polyols.

**Table 3 nutrients-16-03596-t003:** Themes, codes and quotes on barriers in low FODMAP diet delivery.

Themes	Codes	Quotes	Number of Mentions (%)
Challenges related to current practice			
Lack of culturally relevant educational materials	Limited number and access to educational materials	*-“Since I couldn’t find local data, I decided to create my own slides about the FODMAP * foods” (D6)* *-“I don’t have any pamphlet for FODMAPs *, so I write (the information) on a paper.”(D8)* *-“No, we don’t have the list; I just tell them, and they jot down the information.” (D10)*	5 (45.5)
	Limited cultural-based module	*-“The information is from Monash University, which is from overseas, so most of the food listed is Western; it’s not as common as Malaysian cuisine… Malaysian food is less represented.”(D7)* *-“The foods available in the apps are sometimes not suitable for our population…I don’t think the apps is practical for patients to use as references” (D1)* *-“I find it (information available online) quite limited, especially for local food.” (D2)*	7 (63.6)
Limited knowledge about the LFD		*-“ I think there is a limitation in terms of the knowledge of the dietitians in our country. I don’t feel confident to prescribe the diet because my knowledge and training in that area are not sufficient, despite the input I have received until now.” (D2)* *-“I feel like I’m not really qualified to give this advice. I need to pass the Monash test first before providing my clients with those treatments” (D8)* *-“We did not know about FODMAPs * until I attended a talk about it. From the talk, I learned about FODMAPs *.” (D10)*	3 (27.3)
Inadequate formal training among dietitians	Lack of training	*-“Even until now, I feel that I still lack training in this area” (D2)* *-“I hope there are affordable courses offered for dietitians. In our degree’s course, there is not enough time to have exposure to IBS management.”(D8)*	7 (63.6)
Lack of integration in multi-disciplinary care	Low referral rates from doctors or gastroenterologists	*-“(Referrals are) not that often, honestly. It depends on the doctors. If the gastroenterologist is not really (familiar with the) diet, we won’t get the referral unless the patient asks.” (D6)* *-”The main issue, I think, is that doctors do not refer patients to us. We are the last resource that the consultant will ask for help.” (D10)* *-”We don’t normally see their progression, because the doctor didn’t refer them back to us.” (D3)*	2 (18.2)
Challenges in patient management			
Low health literacy of patients	Low and medium education level	*-“I think the education level also plays an important role here”(D3)* *-“This hospital population is not highly educated, so it is very difficult for us to ensure that the patients write down everything they consume (in food diary).” (D11)*	3 (27.3)
Low compliance rate among patients	Refuse to make commitment to comply and bear the symptoms	*-“Patients’ willingness to follow is also one of the challenges. (I mean) patient’s willingness to bear the symptoms (during reintroduction phase, let’s say if they still have bloating with certain foods.” (D5)* *-“They’re not very willing to go into details when it comes to adhering to the elimination phase because it requires a lot of commitment” (D2)*	5 (45.5)
	Refuse to make lifestyle changes	*-“ They told me that this diet is inconvenient for them, or their family member can’t follow it with them” (D6)* *-“Patients always feel like this is burdensome for them to restrict because most of the high FODMAPs * foods are common in our surroundings” (D3)*	4 (36.4)
	Refuse to follow up	*-”They often don’t come back for follow-up, especially during the reintroduction phase, so I don’t know the outcome or their situation.” (D3)* *-”In my opinion, general public still doesn’t fully understand the value of dietitians; Patients seldom come back for follow-up.” (D8)* *-“In my experience, I have very few opportunities to see patients again (because) patients did not return for follow-ups.” (D9)*	3 (27.3)
	Eating out	*-“When they eat out, it’s very difficult to actually control the ingredients.” (D2)* *-“We do not have any product that are FODMAPs * friendly or kind of low FODMAPs * menu in our restaurant, so it’s very hard to comply if the patients eat outside” (D1)*	4 (36.4)
Restrictions for certain populations	Vegetarians	*-“If the patient is a vegetarian, their food choices are already limited. If they follow this diet, the variety of food will be much more restricted, leaving them with fewer options, such as avoiding legumes and lactose.” (D5)*	2 (18.2)
	Pregnant mothers	*-“If the patient is pregnant and needs calcium, it becomes difficult to meet the requirement while following this diet and gaining weight.” (D5)*	1 (9)
	Students	*“If the patient is a student, there is a limitation of food availability. Since they cannot cook for themselves as they stay in the hostel, they are left with no choice but to buy food from the food court or cafes, which may lead to them consuming foods with high FODMAPs *” (D5)*	2 (18.2)

* FODMAPs: Fermentable oligo-, di-, mono-saccharides and polyols.

## Data Availability

The original contributions presented in the study are included in the article and Appendix A, further inquiries can be directed to the corresponding author.

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
