# Peer review of "Practices and Barriers in Implementing the Low FODMAP Diet for Irritable Bowel Syndrome Among Malaysian Dietitians: A Qualitative Study"

_nutrients, 2024, doi:10.3390/nu16213596_

Round 1

Reviewer 1 Report

Comments and Suggestions for Authors

 Practices and Barriers in Implementing the Low FODMAP Diet for Irritable Bowel Syndrome among Malaysian Dietitians: A Qualitative Study

 Major comments:

The authors suggested that the practice among dietitians in Malaysia is not standardized and most dietitians were not able to provide dietary advice (for Irritable Bowel Syndrome patients) based on evidence-based practice (because of lack of standardization in the practices and culturally specific resources, inadequate training and knowledge gap among dietitians).

Results were given rather clearly with sufficient tables and data analysis. Studies are based on recent literature (24 items from 56 were published in the last five years).

 Particularly alarming is the underestimation of the low FODMAP (Fermentable Oligo-, Di-, Mono-saccharides And Polyols) diet for IBS patients in the supportive therapy of patients, as evidenced by the sentence:

‘A participant also mentioned that doctors primarily rely on medication and consider diet as a last resort.’

 An important paragraph at the paper's end (in 4.3 Strengths and limitations SECTION) includes LIMITATIONS OF THE STUDY. Please introduce limitations connected with small male dietitians in the study group.

Minor comments:

 Instead of:

Meanwhile, seven barriers were identified: 1. lack of cultural relevant educational materials; 2. knowledge gap among dietitians; 3. inadequate formal training among dietitians; 4. lack of integration in multi-disciplinary care; 5. low health literacy of patients 6. low compliance rate among patients and 7. restrictive for certain populations.

Should be:

Meanwhile, seven barriers were identified: 1. lack of culturally relevant educational materials; 2. knowledge gap among dietitians; 3. inadequate formal training among dietitians; 4. lack of integration in multi-disciplinary care; 5. low health literacy of patients; 6. low compliance rate among patients, and 7. restrictive for certain populations.

Instead of:

Malay, Chinese, Indian and East Malaysian foods make up the Malaysian diet, which is high in FODMAPs, as mainly from onion, garlic and shallots used in cooking [10], tropical fruits that high in fructose [11], as well as refined wheat products [12].

Should be:

Malay, Chinese, Indian and East Malaysian foods make up the Malaysian diet, which is high in FODMAPs, mainly from onion, garlic and shallots used in cooking [10], tropical fruits that are high in fructose [11], as well as refined wheat products [12].

Instead of:

-“ I usually tell my patients that if they really want to see the difference during this whole programme, they will need to minimise eating out and prepare food from home” (D2)

Should be:

-“ I usually tell my patients that if they really want to see the difference during this whole programme is that they will need to minimise eating out and prepare food from home” (D2)

Instead of:

- “My checklist is from Monash university and King's college London KCL” (D1)

Should be:

- “My checklist is from Monash University and King's College London KCL” (D1)

Instead of:

A number of barriers were reported in detail by the participants and were classified into two categories: (i) challenges related to current practice and (ii) challenges in patient management (Table 3).

Should be:

A few barriers were reported in detail by the participants and were classified into two categories: (i) challenges related to current practice and (ii) challenges in patient management (Table 3).

Instead of:

In the area of challenges related to current practice, four main themes emerged: (i) lack of cultural relevant educational materials, (ii) knowledge gap among dietitians, (iii) inadequate formal training among dietitians, and (iv) lack of integration in multi-disciplinary care.

Should be:

In the area of challenges related to current practice, four main themes emerged: (i) lack of culturally relevant educational materials, (ii) knowledge gap among dietitians, (iii) inadequate formal training among dietitians, and (iv) lack of integration in multi-disciplinary care.

Instead of:

-“Since I couldn't find local data, decided to create my own slides about the FODMAPs foods” (D6)

Should be:

-“Since I couldn't find local data, I decided to create my own slides about the FODMAPs foods” (D6)

Instead of:

-“This hospital population is not highly educated, so it is very difficult for us to ensure that the patient write down everything they consume (in food diary).” (D11)

Should be:

-“This hospital population is not highly educated, so it is very difficult for us to ensure that the patient writes down everything they consume (in food diary).” (D11)

Comments on the Quality of English Language

 Minor editing of English language required.

Reviewer 2 Report

Comments and Suggestions for Authors

The low FODMAP diet is a dietary intervention for IBS patients, but its application among Malaysian patients is not well understood. This study explores the practices and barriers to delivering the low FODMAP diet among Malaysian dietitians. In-depth interviews with practising dietitians uncovered seven significant challenges in effectively delivering dietary guidelines. These challenges included the absence of standardized implementation procedures, insufficient training opportunities, the lack of culturally specific resources, etc. It was suggested that training programs and resource development are needed to support Malaysian dietitians in managing IBS patients.

A few suggestions: 

1. Please describe the regions and education levels of the participating dietitians,  any training program received, and whether these participating dietitians are representative nationwide in Malaysia.

2. Nine out of the eleven participating dietitians are female. This ratio might not be representative nationwide in Malaysia and could indicate bias in data collection. This limitation needs to be added to the discussion.

3. Please delete the Journal instruction sentence at the beginning of the Results section.

4. Please put the Interview questionnaire as the Supplementary materials. 

5. Can any interview questions and answers can be analysed quantitatively?

6. Please investigate if there are any dietitian training programs, both online or in person, in Malaysia and what the requirements are to become a qualified dietitian. This information needs to be added in the Introduction section. 

Reviewer 3 Report

Comments and Suggestions for Authors

This manuscript describes the qualitative study the investigators completed to learn about the practices and barriers to implementing the low FODMAP diet in Malaysia.

·        Abstract: on line 28, the second barrier is the knowledge gap among dietitians. Can you be more specific?

·        Line 233: Is “among dietitians” referring to the dietitians in the study or dietitians in general?

·        Lines 245 to 256 seem to be a summary of the literature. Please relate it to this study.

·        Lines 88 to 98 describe the study population. The authors state there are few dietitians in Malaysia who specialize in gastrointestinal counseling. How many RDs are in Malaysia? What was the sample size you had to select from? Several recruiting methods were used. Which one was the most successful?

Comments on the Quality of English Language

Overall, the English language is good. It should be checked for grammatical errors, and there are a few parts where the sentence structure could be improved.
